# Studies on Carrier Recombination in GaN/AlN Quantum Dots in Nanowires with a Core–Shell Structure

**DOI:** 10.3390/nano10112299

**Published:** 2020-11-20

**Authors:** Jun Deng, Zhibiao Hao, Lai Wang, Jiadong Yu, Jian Wang, Changzheng Sun, Yanjun Han, Bing Xiong, Hongtao Li, Wei Zhao, Xihui Liang, Junjun Wang, Yi Luo

**Affiliations:** 1Department of Electronic Engineering, Beijing National Research Center for Information Science and Technology, Tsinghua University, Beijing 100084, China; dengjun16@mails.tsinghua.edu.cn (J.D.); wanglai@tsinghua.edu.cn (L.W.); yjd13@tsinghua.org.cn (J.Y.); wangjian@tsinghua.edu.cn (J.W.); czsun@tsinghua.edu.cn (C.S.); yjhan@mail.tsinghua.edu.cn (Y.H.); bxiong@mail.tsinghua.edu.cn (B.X.); lihongtao@tsinghua.edu.cn (H.L.); luoy@tsinghua.edu.cn (Y.L.); 2Institute of Semiconductors, Guangdong Academy of Sciences, Guangzhou 510070, China; weizhao@gdisit.com (W.Z.); liangxh@hotmial.com (X.L.); junjunwang@gdisit.com (J.W.)

**Keywords:** GaN/AlN, quantum dots, nanowire, core–shell

## Abstract

GaN quantum dots embedded in nanowires have attracted much attention due to their superior optical properties. However, due to the large surface-to-volume ratio of the nanowire, the impacts of surface states are the primary issue responsible for the degradation of internal quantum efficiency (IQE) in heterostructured dot-in-nanowires. In this paper, we investigate the carrier recombination mechanism of GaN/AlN dot-in-nanowires with an in situ grown AlN shell structure. Ultraviolet photoelectron spectroscopy (UPS) measurements were performed to describe the band bending effect on samples with different shell thicknesses. Temperature-dependent photoluminescence (TDPL) data support that increasing the AlN shell thickness is an efficient way to improve internal quantum efficiency. Detailed carrier dynamics was analyzed and combined with time-resolved photoluminescence (TRPL). The experimental data are consistent with our physical model that the AlN shell can effectively flatten the band bending near the surface and isolate the surface non-radiative recombination center. Our systematic research on GaN/AlN quantum dots in nanowires with a core–shell structure may significantly advance the development of a broad range of nanowire-based optoelectronic devices.

## 1. Introduction

Over the past decade, III-nitride semiconductor quantum dots in nanowires have attracted much attention due to their potential applications in solid-state lighting [1,2,3,4], photochemical sensing [5], and quantum cryptography [6]. In particular, several groups have reported self-assembled III-nitride quantum dots in nanowires grown by molecular beam epitaxy (MBE) [7,8,9,10]. Due to the large surface-to-volume ratio, nitride-based nanowires exhibit high crystalline quality by strain relaxation [10]. Moreover, III-nitride quantum disks embedded in nanowires demonstrate superior optical properties, such as stronger quantum confinement [6,11], smaller Auger recombination coefficient [12], and suppressed quantum-confined Stark effect (QCSE) [1]. Thus, the dot-in-wire structure is more flexible when compared to the planar quantum well.

However, carrier dynamics have been adversely affected by inevitable surface recombination caused by surface states due to the large free surface [13]. These surface states also lead to Fermi level pinning, which is responsible for the depletion of charge carriers [14,15]. Both of them are the main factors causing the performance degradation of low-dimensional devices. In this regard, various surface passivation techniques have been explored. Surface cleaning via diluted potassium hydroxide [16], ammonia [17], or sulfides [18,19] is commonly used to improve quantum efficiency by reducing surface states and removing surface oxidation. In addition, the deposition of dielectric films, such as SiO_2_, SiN_x_ and Al_2_O_3_, is a widely used passivation method for the passivation of GaN thin film surfaces [20,21]. Owing to the characteristics of shape retention of atomic layer deposition, dielectric deposition can also be applied to nanowires to suppress the band-bending effect caused by surface states [13]. All of these passivation approaches have demonstrated effectiveness in improving device performance but require further treatments after the epitaxy. In situ deposition of wide-bandgap semiconductors on the sidewalls of nanowires to form core–shell structures has been shown to be highly promising for improving the performance of nanowire quantum dot devices [22,23]. However, the physical mechanism of passivation by a large bandgap shell has yet to be comprehensively understood.

In this work, we investigated the carrier recombination mechanism of GaN/AlN quantum dots in nanowires grown on silicon substrates using MBE. By precisely adjusting the Al flux, GaN/AlN quantum dots in nanowires with different AlN shell thicknesses were obtained. Ultraviolet photoemission spectroscopy (UPS) was employed to quantitatively analyze the surface binding energy shift of different samples. The carrier dynamics were further studied by measurements of temperature-dependent photoluminescence (TDPL) and time-resolved photoluminescence (TRPL). The results reveal that, by increasing the thickness of the AlN shell, the band bending of GaN/AlN quantum dots in nanowires is substantially released, which leads to an increase of the overlap of electron–hole wave functions and the radiation recombination rate. Furthermore, due to the strong isolation of the AlN shell barrier, the carriers in GaN quantum dots can be protected from being captured by the non-radiative recombination centers on the sidewalls. The measurements of surface potential and carrier dynamics provide a complete perspective for the understanding of the self-passivation mechanism of core–shell structured GaN/AlN quantum dots in nanowires.

## 2. Materials and Methods

### 2.1. Preparation of GaN/AlN Quantum Dots in Nanowires

The self-assembled formation process of the GaN/AlN dot-in-nanowire core–shell heterostructures is schematically illustrated in Figure 1. GaN nanowires, with a height of 200 nm, are grown on (111) Si substrates after nitridation at 760 °C, as shown in Figure 1a. As illustrated in Figure 1b, the AlN barrier subsequently grown at 890 °C covers the top rather than the sidewall of the GaN nanowire due to the high migration ability of Al atoms at high temperatures. The GaN quantum dot, shown in Figure 1c, with a height of 1–2 nm, is grown at the center of the nanowire. The AlN cap grown at 760 °C can cover the growth front, including the top and sidewall of the GaN quantum dot as shown in Figure 1d, which can be explained by the low migration ability of Al atoms at a low temperature [24,25]. Therefore, the spontaneous wide-bandgap shell is formed. Typical clustered and dispersed GaN/AlN dot-in-nanowire samples are shown in Figure 1e,f. Samples with different shell thicknesses can be obtained by deposition under various Al beam equivalent pressures (BEP) including 4.35 × 10^−8^, 8.1 × 10^−8^, 1.02 × 10^−7^, 1.5 × 10^−7^, 2.7 × 10^−7^ Torr.

### 2.2. Characterizations

Field emission scanning electron microscopy (FESEM, Zeiss, Merlin, Jena, Germany, at 5 kV applied voltage) and transmission electron microscopy (TEM, JEOL, JEM-2100, Tokyo, Japan, at 200 kV acceleration voltage) were applied to characterize the crystalline structures and morphologies of the samples. UPS, integrated on an X-ray photoelectron spectroscopy (XPS, Kratos, AXIS ULTRA DLD, Manchester, UK), was used to measure the surface potential which can be deduced by parallel shifts of binding energy. Emitted electrons were excited by irradiated HeI and the signal scanning range was −2 eV to 23 eV. The signal of TDPL was collected by a monochromator (Jobin Yvon 550, Paris, France) with a resolution of 0.5 nm. A lifetime spectrometer (Hamamatsu, Hamamatsu, Japan) and a third-harmonic laser (Spectra Physics, Santa Clara, CA, USA) with a wavelength of 266 nm, a pulse width of 170 fs, and a repetition rate of 8 MHz were employed to record TRPL decay of the samples.

## 3. Results and Discussion

The samples used for comparison experiments and the related parameters are summarized in Table 1. Samples a–d were employed to explore the thickness of the AlN shell under various Al BEPs. GaN quantum dots embedded in these samples were grown for 60 s to facilitate the observation of the interface between the shell and the dot. A detailed view in Figure 2a–d shows that GaN dots are capped by the AlN barriers, each of which forms an AlN shell at the nanowire sidewall. No defects are observed at the interfaces between the GaN quantum dots and the AlN shells. Therefore, only surface states on AlN shells are considered in the following discussion.

From samples a–d, it can be seen that the shell thickness is directly correlated to the Al BEP. Previous studies related to AlN growth have shown the same tendency [24]. Samples e–g were employed to study the carrier recombination mechanism, and their shell thicknesses can be estimated according to the relationship between Al BEP and AlN shell thickness illustrated in Figure 2e. Ultra-thin GaN quantum disks (around 1 nm thick) were grown in these samples for high carrier confinement to distinguish spectra from the emission of GaN nanowire base. The high resolution transmission electron microscope (HRTEM) image in Figure 3a reveals that the ultra-thin GaN disk is free from threading dislocation. The selected area electron diffraction (SAED) pattern in Figure 3b shows that the nanowires have a hexagonal crystal structure without any other growth phase and stacking faults. The extra spots along the c-axis indicate the difference of the lattice parameter of GaN and AlN along the growth direction. It is worth noting that there is no additional spot separation along the direction perpendicular to the c-axis, which indicates the same lattice parameter of GaN and AlN perpendicular to the c-axis, i.e., the fully compressive strain of GaN quantum disks.

The band structure near the sidewall is altered after depositing an AlN shell on the sidewall of the nanowire. By measuring the parallel shift of the binding energies, the surface potential can be quantitatively measured with UPS as the thickness of the AlN shell changes. Figure 4a shows the energy distributions of the emitted electrons from the samples e–g. The left part of the spectra provides the relative positions of valence band maximum (VBM) with the Fermi level (E_f_) as a reference, which can be deduced through the intersections of the reverse extension lines of the uplift part of the spectra and the horizontal dashed lines [26]. The VBM energies (E_v_) of different samples have different degrees of energy shifts. For sample e, the E_v_ is located at 5.599 eV below E_f_. Considering the bandgap of AlN bulk material, it can be inferred that the valence band near the edge bends upward along the radial direction. As the thickness of the AlN shell increases, E_f_–E_v_ also increases. For sample g, which has the thickest AlN shell, E_f_–E_v_ is about 0.44 eV larger than that of sample e, which can be attributed to the apparent flattening of the upward band bending near the AlN sidewall. For ideal semiconductor surfaces, the dangling bond density is constant. While the nanowires with larger diameters are grown under higher Al BEPs, which may reduce N deficiency defects. Thus, the band bending can be adjusted for nanowires with different diameters through regulating the surface states. It is also noted that the VBM energy displacement of AlN nanowires is significantly smaller than that of GaN nanowires, which also shows the superiority of the AlN shell for passivation.

Figure 4b shows the schematic diagram of the energy band, which illustrates the band bending with different shell thicknesses. The large concentration of surface states can regulate the surface electrical properties, including space-charge related surface depletion, Fermi level pinning at the sidewall of nanowires, and upward band bending [14,15]. For quantum dot-in-nanowire samples with larger diameters, such as the etched planar quantum well, only the peripheral part near the sidewall is depleted, while the physical properties of the central part of the nanowire are consistent with the bulk material [27]. In this case, the radiative recombination mostly occurs at the inner parts of the quantum dots. Thus, the effects of band bending can be ignored, while the surface states act as nonradiative recombination centers. However, for self-assembled GaN nanowires with diameters of no more than 80 nm, the inner parts of the nanowires are completely depleted [14]. The band bending causes holes to accumulate at the edge of the quantum dot, while electrons prefer the core of the column. This will lead to a decrease in the overlap of the wave function of the electrons and holes, which may reduce the radiative recombination rate of carriers. Moreover, the increase of the diameter of fully depleted nanowires will cause a decrease of the electron energy in the core, thereby causing the red shift of the emission peak.

As for the AlN nanowires with an average diameter of less than 60 nm applied in this paper, the depletion region near the nanowire sidewall cannot be ignored. Compared to GaN nanowires, the lower surface potential of AlN nanowires means a narrower depletion region, which causes partial depletion of the nanowire. The depletion region extends to the edge of the GaN quantum dot, while the center of the quantum dot is most probably not depleted. In the meantime, as the thickness of the AlN shell increases, the depletion region becomes narrower. This can be explained by the fact that the amount of surface charge is linearly proportional to the nanowire radius, whereas the amount of body charge is proportional to the square of the nanowire radius [28]. Therefore, as the thickness of the AlN shell increases, the depletion region is further drawn away from the quantum dot, and the influence on the edge of the quantum dot is less significant. If the shell thickness further increases, the depletion region will completely shift out of the quantum dot, and the improvement of the carrier recombination rate in the quantum dot tends to be saturated. Thus, the surface states can be effectively isolated using an AlN shell in a GaN/AlN dot-in-nanowire, as in Figure 4b, through suppressing the separation of the electrons and holes due to band bending. On the other hand, considering the bandgap of AlN of 6.2 eV, the carriers in the quantum dot have a low probability to pass through the AlN barrier and be captured by the surface non-radiative recombination centers. These are the main physical mechanisms that the AlN shell uses to improves the quantum efficiency of GaN/AlN quantum dots in nanowires. A detailed analysis is provided through the following experiments.

To further illustrate the influence of the AlN shell on the energy band and the surface recombination, the optical properties of the samples mentioned above were systematically studied. Figure 5a shows the typical PL spectrum of sample e with peak analysis measured at room temperature. Via this fitting method, the PL spectra at 8 K and 300 K of the GaN quantum dots obtained by Gaussian fitting are illustrated in Figure 5b. The energetic positions of the peak maxima of the respective samples are a significantly blue shift compared to the relaxed GaN bulk material (365 nm). This can be explained by both the strong quantum confinement effect of GaN quantum dots and the bandgap enlargement of GaN due to the compressive strain exerted by the AlN shield. Despite the 2% compressive stress of GaN dots on the AlN, which causes the strong piezoelectric polarization field, the separation of electrons and holes along the c-axis is suppressed due to the ultra-thin GaN disk. Therefore, the red shift caused by QCSE is not obvious, and the effects of the quantum confinement effect and strain are more significant. Since the growth time of the quantum dots of samples e–g is the same, it was expected that the luminescence peak positions would be the same. However, when increasing AlN shell thickness, the emission peak position changes from 327 nm to 316 nm at 300 K. Meanwhile, the significant blue shift indicates that the AlN shell has a straightforward effect on the band structure of quantum dots, which is consistent with the UPS test results. The internal quantum efficiency (IQE), defined as the ratio of integrated PL intensity at 300 K to that at 8 K, assuming the nonradiative recombination centers are frozen at 8 K, is employed to evaluate the emission properties of the quantum dots [29,30]. The PL intensities at room temperature of all the samples remain at 20% to 40% of their maximum values at low temperatures. This behavior is the characteristic of GaN/AlN nanostructures with a strong 3D confinement, in contrast to the planar quantum well structures which generally exhibit a PL quenching of several orders of magnitude when increasing the measurement temperature [31,32,33]. Monotonically increasing IQE with increasing AlN shell thickness indicates a successful manipulation to the surface states.

We further performed TRPL measurements at the wavelength of the PL peak position to investigate the impact of the AlN shell on the recombination mechanism of the GaN/AlN dot-in-wire structure. The peak intensity of the pulsed excitation was set at 1.59kW/cm^2^. The excitation condition was under a low excitation regime in which the effect of band-bending screening by the photocarriers could be ignored [34]. As plotted in Figure 6, the TRPL results of samples e–g can be well described by a standard stretched exponential model [35],
(1)$$I\left(t\right)=I\left(0\right)e^{-{\left(\frac{t}{\tau }\right)}^{\beta }},$$
where *I*(*t*) denotes the time-varying PL intensity, *τ* is the carrier lifetime of the excited sample, and *β* is the stretching parameter, suggesting that the carrier dynamics of the GaN/AlN quantum dots in nanowires with a core–shell structure are heavily disordered due to the energy band fluctuation [36]. The radiative recombination (*τ_r_*) and nonradiative recombination (*τ_nr_*) lifetimes can be derived from the total decay time and IQE (*η_int_*) listed in Table 2 using the following relations [37],
(2)$$\frac{1}{\tau }=\frac{1}{\tau_{r}}+\frac{1}{\tau_{nr}},$$
(3)$$\eta_{int}=\frac{\tau_{nr}}{\tau_{r}+\tau_{nr}},$$

In the quantum dots, the nonradiative decay is mainly determined by surface recombination and volume recombination. Comparing samples e and f, the nonradiative recombination lifetime increases slightly as the AlN shell becomes thicker. However, further increasing the shell thickness will not improve the nonradiative decay. This can be explained by the lateral confinement potential provided by the AlN shell. Because of its wide bandgap, a relatively thin AlN shell is sufficient to prevent carriers from reaching the detrimental sidewall surface by tunneling decay and thermal decay. The radiative decay is mainly affected by the band bending of the GaN quantum dots. The radiative recombination lifetime becomes shorter as the shell thickness increases, which can be attributed to the increase of overlapping between electron and hole wave functions after band flattening. Thus, adjusting the thickness of the AlN shell is an effective way to manipulate the band bending. Two mechanisms mentioned above lead to the improvement of IQE.

## 4. Conclusions

In this work, hetero-epitaxial GaN/AlN quantum dots in nanowires were grown with an AlN shell, and the carrier recombination dynamics were carefully investigated. TDPL measurements of quantum dots in nanowires with different AlN shell thicknesses on the same core sample show a significant blue shift of PL peak position and improvement of IQE with increasing shell thickness. Combined with UPS and TRPL measurements, the physical nature of carrier recombination in GaN/AlN quantum dots in nanowires with AlN shells can be comprehensively understood in that the AlN shell can effectively release the band bending effect and isolate the surface non-radiative recombination centers. We believe our work may overcome some of the critical challenges of low dimensional materials and advance the development of nanowire-based optoelectronic devices.

## Figures and Tables

**Figure 1 nanomaterials-10-02299-f001:**
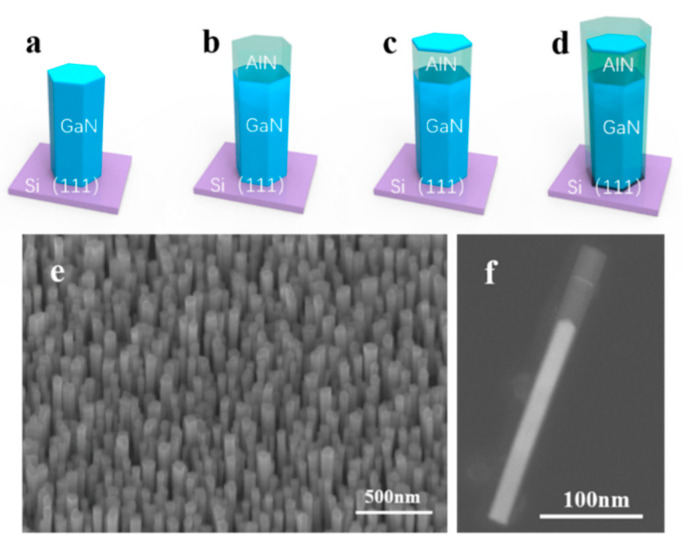
(**a**–**d**) Schematic illustration of the epitaxial growth of catalyst-free GaN/AlN dot-in-nanowire core–shell structure on a Si substrate. (**a**) GaN nanowire template grown on Si(111). (**b**) Formation of the high-temperature grown AlN barrier. (**c**) GaN dot with a thickness of 1–2 nm grown on AlN. (**d**) Formation of the low-temperature grown AlN barrier and shell. (**e**) A 45° tilted SEM image of a typical GaN/AlN dot-in-nanowire sample. (**f**) High-angle annular dark field scanning transmission electron microscopy (HAADF-STEM) image of a typical single nanowire.

**Figure 2 nanomaterials-10-02299-f002:**
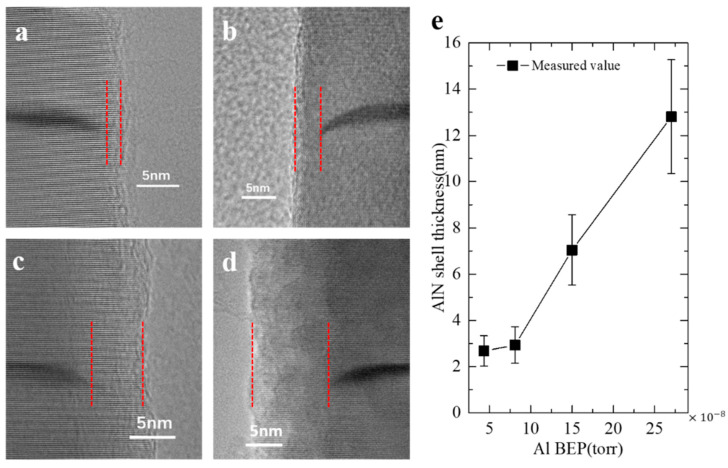
(**a**–**d**) TEM images of AlN shells grown under different Al beam equivalent pressures (BEPs) in samples a–d. The boundaries of the AlN shells are marked by red dotted lines. (**e**) Measured AlN shell thicknesses versus Al BEP. The error bar represents the standard deviation of uncertainty.

**Figure 3 nanomaterials-10-02299-f003:**
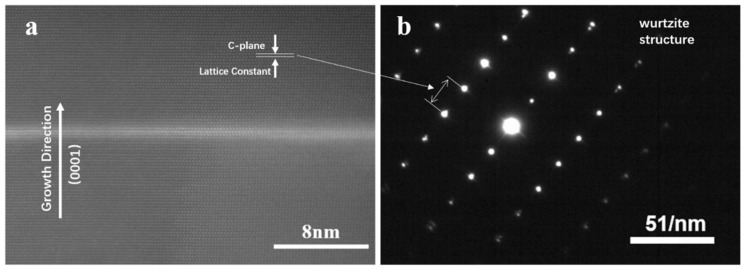
(**a**) HAADF-STEM image of the GaN quantum disk in a single nanowire. (**b**) Selected area electron diffraction (SAED) of the corresponding area of (**a**).

**Figure 4 nanomaterials-10-02299-f004:**
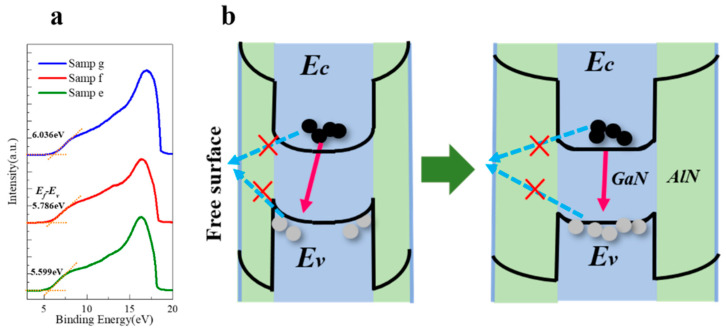
(**a**) Ultraviolet photoelectron spectroscopy (UPS) spectra of the samples with different AlN shell thicknesses. The spectra have been vertically shifted for clarity. (**b**) Schematics of the band structures and distributions of electrons and holes for samples with different AlN shell thicknesses. Arrows indicate the related transitions.

**Figure 5 nanomaterials-10-02299-f005:**
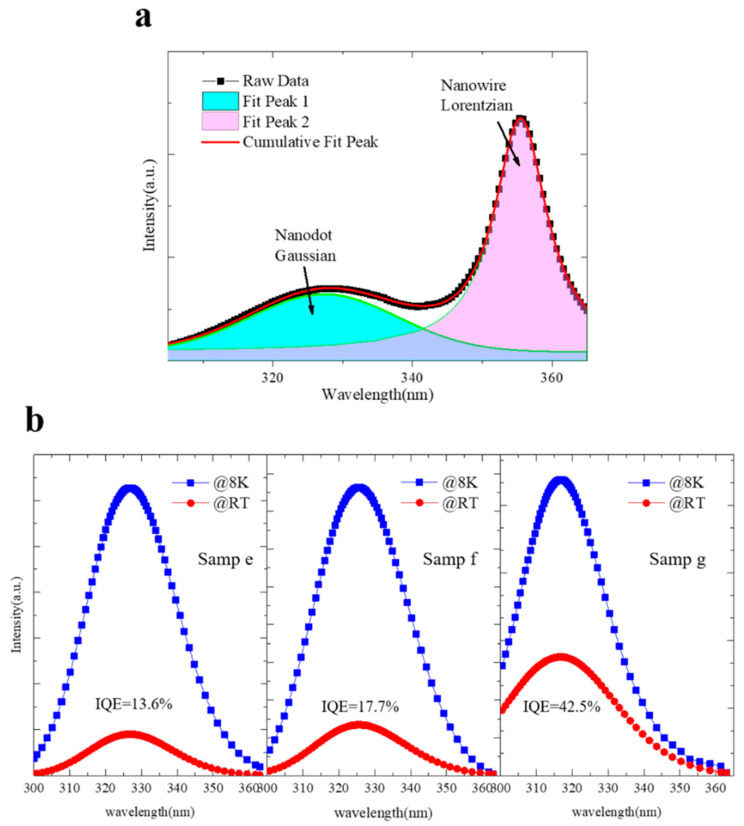
PL results of various samples. (**a**) Typical fitted result of room temperature (RT) PL emissions of sample e. The blue and pink enclosed regions represent the emission of quantum dots and nanowires, respectively. (**b**) Low temperature (8 K) and room temperature PL spectra of GaN/AlN quantum dots in nanowires obtained by Gaussian fit.

**Figure 6 nanomaterials-10-02299-f006:**
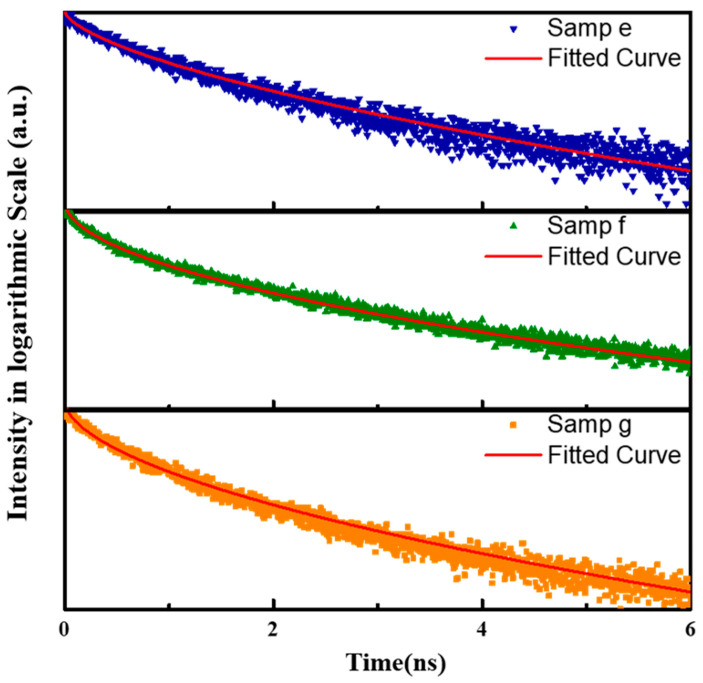
The radiative decay of samples **e**–**g** measured at 300 K with 266 nm of excitation. The red lines represent the fitted curve using a stretched exponential model.

**Table 1 nanomaterials-10-02299-t001:** Summary of the samples used for comparison experiments and the related parameters.

Sample	Al BEP (Torr)	GaN Dot Growth Time (s)	Shell Thickness (nm)
a	4.35 × 10^−8^	60	2.68
b	8.1 × 10^−8^	60	2.93
c	1.5 × 10^−7^	60	7.04
d	2.7 × 10^−7^	60	12.81
e	8.1 × 10^−8^	30	2.93 (estimated)
f	1.02 × 10^−7^	30	4.22 (estimated)
g	1.5 × 10^−7^	30	7.04 (estimated)

**Table 2 nanomaterials-10-02299-t002:** Variation of radiative, nonradiative, and total carrier lifetime in samples e–g fitted by a stretched exponential model.

Sample	*τ* (ps)	*τ_nr_* (ps)	*τ_r_* (ps)
e	817	946	6007
f	903	1097	5103
g	613	1066	1443

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
