# Peer review of "Studies on Carrier Recombination in GaN/AlN Quantum Dots in Nanowires with a Core–Shell Structure"

_nanomaterials, 2020, doi:10.3390/nano10112299_

Round 1
Reviewer 1 Report
In this paper, the authors investigate the recombination mechanisms in order to improve the internal quantum efficiency (IQE) of the GaN/AlN quantum dot embedded in a nano-wire. By controlling the AlN shell thickness it is found that IQE can be improved substantially. They also showed the agreement with the physical model simulation and the experimental results. I would say the results warrant publication. Nevertheless, the paper can be improved by including a little more detailed discussion of the physical model employed such as the band structure, optical model, etc.
Reviewer 2 Report
The article reports on a systematic investigation of GaN core-shell nanowires containing a single GaN/AlN quantum dot and more precisely on the effect of an AlN shell on the carrier recombination. In addition to structural characterizations, the study relies on ultraviolet electron spectroscopy for assessing the Fermi energy with respect to the valence band, temperature-dependent photoluminescence spectroscopy for estimating the internal quantum efficiency as well as time resolved photoluminescence for estimating the radiative and non-radiative recombination rates. This combination of techniques proves to be an excellent approach for disentangling the band-bending and recombination issues. The results reported in this article are quite convincing. This well written and organized paper deserves publication in Nanomaterials providing that the following points be addressed.
- Lines 191-192: the strain exerted by the AlN shield on the nanowire core could also explain the PL energy shift. The strain issue must be discussed.
- For the TRPL experiments, the peak intensity of the pulsed excitation must be provided and the effect of band-bending screening by the photocarriers, which has been thoroughly investigated in the literature, must be discussed.
- Figure 6: it is preferable to plot the decay using a logarithmic scale.
- Is the decay mono- or multi- exponential? This information is not provided.
Typos:
- Line 113: Figure 2e (and not 2f).
- Line 176: “are with low probability” have a low probability.
Round 2
Reviewer 2 Report
The revised version of the manuscript is now ready for publication. All comments have been addressed thoroughly and properly.